# Trial Design of a Prospective Multicenter Diagnostic Accuracy Study of a Point-of-Care Test for the Detection of *Taenia solium* Taeniosis and Neurocysticercosis in Hospital-Based Settings in Tanzania

**DOI:** 10.3390/diagnostics11091528

**Published:** 2021-08-24

**Authors:** Chiara Trevisan, Inge Van Damme, Bernard Ngowi, Veronika Schmidt, Dominik Stelzle, Karen Schou Møller, Mwemezi Kabululu, Charles E. Makasi, Pascal Magnussen, Emmanuel Bottieau, Emmanuel Abatih, Maria V. Johansen, Helena Ngowi, Benedict Ndawi, Kabemba E. Mwape, Gideon Zulu, Pierre Dorny, Andrea S. Winkler, Sarah Gabriël

**Affiliations:** 1Department of Biomedical Sciences, Institute of Tropical Medicine, 2000 Antwerp, Belgium; pdorny@itg.be; 2Department of Veterinary Public Health and Food Safety, Faculty of Veterinary Medicine, Ghent University, 9820 Merelbeke, Belgium; inge.vandamme@ugent.be; 3National Institute for Medical Research, Muhimbili Medical Research Centre Dar es Salaam, Dar es Salaam 11101, Tanzania; b_ngowi@yahoo.co.uk (B.N.); charlesmakasi2021@gmail.com (C.E.M.); 4Department of Public Health, University of Dar es Salaam Tanzania, Dar es Salaam 65015, Tanzania; 5Department of Neurology, Center for Global Health, Klinikum rechts der Isar, Technical University of Munich, 81675 Munich, Germany; veronika.schmidt@tum.de (V.S.); dominik.stelzle@tum.de (D.S.); andrea.winkler@tum.de (A.S.W.); 6Centre for Global Health, Institute of Health and Society, Faculty of Medicine, University of Oslo, 0450 Oslo, Norway; 7Chair of Epidemiology, Department of Sport and Health Sciences, Technical University of Munich, 80809 Munich, Germany; 8Department of Veterinary and Animal Sciences, Faculty of Health and Medical Sciences, University of Copenhagen, 1870 Frederiksberg, Denmark; karenm@sund.ku.dk; 9Tanzania Livestock Research Institute (TALIRI)-Uyole, Mbeya P.O. Box 6191, Tanzania; mwemezie@gmail.com; 10Faculty of Health and Medical Sciences, University of Copenhagen, 2200 Copenhagen, Denmark; pma@sund.ku.dk; 11Department of Clinical Sciences, Institute of Tropical Medicine, 2000 Antwerp, Belgium; EBottieau@itg.be; 12Department of Applied Mathematics, Computer Sciences and Statistics, Faculty of Sciences, Ghent University, 9000 Ghent, Belgium; Emmanuel.Abatih@UGent.be; 13Independent Researcher, 2100 Copenhagen, Denmark; mariavangjohansen@gmail.com; 14Department of Veterinary Medicine and Public Health, College of Veterinary Medicine and Biomedical Sciences, Sokoine University of Agriculture, Morogoro P.O. Box 3021, Tanzania; h_ngowi@yahoo.com; 15Primary Health Care Institute, Iringa 235, Tanzania; ndawib@gmail.com; 16Department of Clinical Studies, School of Veterinary Medicine, University of Zambia, Lusaka 10101, Zambia; evans.mwape@unza.zm (K.E.M.); gideonzulu@yahoo.com (G.Z.); 17Eastern Provincial Health Office, Ministry of Health, Lusaka 30205, Zambia

**Keywords:** *Taenia solium*, taeniosis, cysticercosis, hospital-based settings, point-of-care test, rapid tests, diagnostics

## Abstract

*Taenia solium* diagnosis is challenging as trained personnel, good diagnostic tools, and infrastructure is lacking in resource-poor areas. This paper aims to describe the study trial design adopted to evaluate a newly developed rapid point-of-care test that simultaneously detects taeniosis and neurocysticercosis (TS POC) in three district hospitals in Tanzania. The two-stage design included three types of patients: patients with specific neurological signs and symptoms (group 1); patients with complaints compatible with intestinal worm infections (group 2); patients with other symptom(s) (group 3). For group 1, all patients were tested using the TS POC test (stage 1), after which all positive, and a subset of negative, patients were selected for laboratory reference tests, clinical examination, and a brain computed tomography (CT) scan (stage 2). For groups 2 and 3, a similar design was adopted, but clinical examination and a brain CT scan (stage 2) were only performed in patients who were TS POC test-positive for cysticercosis. Due to the lack of a gold standard, a Bayesian approach was used to determine test accuracy for taeniosis and cysticercosis. For neurocysticercosis, a composite case definition was used as the reference standard. If successful, this study will help the future developments (commercialization and implementation) of the rapid test and improve patient management and disease prevention.

## 1. Introduction

*Taenia solium* is a neglected zoonotic parasite contributing to the significant health and economic burden in affected areas of low- and middle-income countries (LMIC) [1].

In Tanzania, several projects on *T. solium* cysticercosis have been conducted and provided evidence of the parasite presence in several regions of the country [2]. Based on antigen enzyme-linked immunosorbent assay (Ag-ELISA) and antibody (Ab)-ELISA, the prevalence of human cysticercosis (CC) was 16.7% and 45.3%, respectively, in the Songwe region, southern Tanzania in 2009. In the same area, the prevalence of taeniosis (T) was estimated to be 5.2% using copro-Ag ELISA [3]. In the Mbulu district, northern Tanzania, a hospital-based brain computed tomography (CT) scan study indicated that almost 20% of people with epilepsy suffered from neurocysticercosis (NCC) [4,5].

At the district hospital level, the diagnosis of tapeworms can be performed using microscopy, but has a low sensitivity and fails to differentiate *Taenia* spp. Other methods, such as the copro-antigen ELISA, the recombinant antigen-based enzyme-linked immunoelectrotransfer blot (EITB), and polymerase chain reaction (PCR) techniques are available but also have limitations in their performance, and are used only in research settings as laboratories at the district hospital level in Tanzania lack equipment, adequately trained experts, and infrastructure [6]. Moreover, differential diagnosis and the stigma associated with epilepsy further contribute to the challenges that district hospitals face [7].

To standardize the diagnosis of NCC, Del Brutto et al. [8] formulated a set of criteria based on neuroimaging, assisted by serological, clinical, and epidemiological results. Neuroimaging is possible using CT and/or magnetic resonance imaging (MRI). For serology, the lentil lectin-bound glycoproteins-enzyme-linked immunoelectrotransfer blot (LLGP-EITB) is the test of choice [8,9]. In resource-poor areas, this is problematic as access to neuroimaging is very limited, trained neurologists are scarce, and the LLGP-EITB, though considered as having high specificity and sensitivity [10], is expensive and in a format not applicable/available in most local laboratories. Therefore, disease diagnosis in resource-poor areas outside a research setting is hampered by the lack of good tools and skilled personnel, as well as the high costs. 

To limit these challenges, a user-friendly, rapid, cheap *T. solium* point-of-care test (TS POC) was developed by the Centre for Disease Control and Prevention, Atlanta, USA, in collaboration with the Klinikum rechts der Isar, Technical University of Munich, Germany. 

The main objective of the SOLID project (“Evaluation of an antibody-detecting point-of-care test for the diagnosis of *Taenia solium* taeniosis and neurocysticercosis in communities and primary care settings of highly endemic, resource-poor areas in Tanzania and Zambia, including the training of—and technology transfer to the Regional Reference Laboratory and health centers”) was to evaluate the sensitivity/specificity and ease of use of the TS POC test for the detection of *T. solium* T and NCC in resource-poor, highly endemic areas in sub-Saharan Africa. The study in Zambia was conducted in a community-based setting and its design was published elsewhere [11]. This paper focuses on the design of the diagnostic accuracy evaluation of the TS POC test at the district hospital level in Tanzania. 

## 2. Materials and Methods

### 2.1. TS POC Test

A detailed description of the TS POC test is presented in [11]. To summarize, the TS POC test is an antibody-detecting prototype developed by a team of researchers from the USA and Germany. The TS POC test consists of a double-strip cassette that holds one strip for the detection of antibodies against the adult stage (using the recombinant T antigen rES33), and one to detect antibodies against the larval stage of the parasite (using the recombinant CC antigen rT24H) [12,13]. Each strip has one test line, one control line, and a separate port for the sample application, enabling the simultaneous detection of T and CC antibodies. 

For the TS POC test evaluation in Tanzania, 3000 tests were produced at CDC Atlanta in collaboration with Arista Biologicals Inc, Allentown, PA, USA. 

Details on how the TS POC test was performed and read can be found in [14]. In essence, two times 20 micro-liters of blood are collected from a fingertip using a micropipette and placed on the sample well for T and CC, respectively. Immediately after, the chase buffer is applied, and as soon as the flow commences the timer is set for 20 min (Figure 1). The TS POC test is read by two independent readers (two-factorial design), and in the case of a disagreement, a third reader intervenes. A result card is used to report the results which are subsequently entered in electronic case report forms (eCRFs). In the case of an invalid result (e.g., no flow, or control line missing), the test is repeated once.

### 2.2. Study Design, Recruitment, and Eligibility

To evaluate the diagnostic performance of the TS POC test for the detection of T, CC, and NCC, a prospective, two-stage, multicenter study was conducted. The TS POC test was evaluated at the stage where it may be implemented in practice, being patients attending district hospitals in *T. solium*-endemic areas. Since the accuracy of the test was likely to vary across subgroups of patients [15], the study was performed on three groups of patients, namely patients with specific neurological signs and symptoms (epilepsy and/or severe progressive chronic headache) (group 1); patients with complaints compatible with intestinal worm infections (group 2); patients with other symptom(s) (group 3). To get a representative sample for the studied populations, subjects for groups 1 and 2 were consecutively recruited, while for group 3, only every tenth patient was included if eligibility criteria were met. Recruitment took place between December 2017 and February 2020. 

Patients of group 1 were recruited from the mental health ward or the outpatient department while patients belonging to groups 2 and 3 were recruited from the outpatient department only. To be included, patients had to be ten years old or above and willing and able to participate in all aspects of the study (including providing blood and stool samples, undergoing a clinical examination, and a brain CT scan, if necessary), willing and able to provide written informed consent (assent for minors with consent from a parent or a legally authorized representative), living in the study area for the past three months and planning to stay in the same area throughout the study period. Pregnant women and patients suffering from severe health conditions needing in-patient care were excluded from the study. 

To be eligible for inclusion in group 1, patients needed to present to the mental health ward or outpatient department with on-going symptoms of severe progressive chronic headache impeding their daily activities without fever and signs of cerebral infection or other obvious causes; and/or they should have a history of one or more epileptic seizures without obvious causes. Participants meeting any of the following criteria were not enrolled in the study: less than ten years of age; acute febrile illness; fever and signs and symptoms for middle ear infection; signs and symptoms of meningitis; neck muscle pain and stiffness; a history of stroke with neurological focal deficit; a history of psychiatric problems.

To be eligible for inclusion in group 2, patients had to present at the outpatient department with intestinal complaints suggestive of an intestinal worm infection (diarrhea and/or abdominal pain/discomfort for more than two weeks and/or a recent history of worm expulsion). For group 3, eligible patients with any other symptom(s) not immediately linked with intestinal worm infections or symptoms described for group 1 were also recruited from the outpatient department.

### 2.3. Study Objectives and Outcomes

The primary objective of the study was to assess the sensitivity and specificity of the TS POC test for the diagnosis of each of the disease presentations (T, CC, and NCC). Due to the lack of a gold standard, three different reference tests were used for the detection of T and CC, respectively (Table 1). The diagnostic accuracy of the TS POC test for the detection of T was determined against two stool-based reference methods (copro antigen ELISA and copro PCR) and a serological test (immunoblot using the recombinant protein rES33). For CC, the reference methods were the EITB-LLGP, rT24H immunoblot, and the B158/B60 monoclonal antibody-based serum antigen ELISA. For NCC, a composite case definition was used as the gold standard based on Del Brutto’s criteria [8]. An overview of the different reference tests included is given in Table 1, and a description of the tests is presented in Section 2.6.

### 2.4. Study Area

The study was conducted in three rural district hospitals in the Songwe and Mbeya regions, Tanzania. District hospitals were included based on the following eligibility criteria: (i) presence of a mental health ward; (ii) proximity to an area where T and NCC were prevalent; (iii) closeness to the Mbeya Radiology Centre equipped with a CT scanner. The three district hospitals in Mbozi (Vwawa), in rural Mbeya (Ifisi), and in Rungwe (Tukuyu) were included (Figure 2).

### 2.5. Study Workflow

After enrollment, patients were subjected to the TS POC test, and based on its results, a subset of participants was subjected to several reference tests, clinical examination, and CT scanning of the brain. Per hospital, approximately one-third of the required sample size was enrolled (see Section 2.11). The study design flow for group 1 is visualized in Figure 3, while Figure 4 shows the study design flow for groups 2 and 3.

### 2.6. Reference Sample Collection, Processing, and Analysis 

Sample collection, processing, shipment, and analysis were the same as for the SOLID project community-based study in Zambia [11]. To summarize, 3 mL of venous blood was collected, and the serum was stored at −20 °C until reference testing using the LLGP-EITB, recombinant immunoblots (rES33 and rT24H), and Ag ELISA. Stool samples were stored at room temperature in 10% formalin for copro-Ag-ELISA, and 70% ethanol for copro PCR. 

The samples were shipped and analyzed in Belgium. Most analyses were performed at the Institute of Tropical Medicine in Antwerp, Belgium, with a 20 percent subset of samples analyzed at Ghent University, Belgium, for quality control. Only the serum samples for the Ag ELISA were fully analyzed at Ghent University, with a 20 percent subset of samples analyzed at the Institute of Tropical Medicine in Antwerp.

In short, the LLGP-EITB was performed as described previously [10,16] with a few modifications. The LLGP-EITB strip was considered positive if at least one out of seven bands were visible. For the recombinant strips, rES33 and rT24H lines for T and CC were combined in one strip. All strips were provided by CDC. The B158/B60 serum antigen ELISA was used as previously described [17]. For the Copro-Ag-ELISA, the procedure was performed as reported previously by [18] and modified by [19]. DNA extraction of stool samples (200 µL in 70% ethanol) was carried out using the QIAmp^®^ fast DNA stool mini kit (Qiagen, Hilden, Germany) according to the manufacturer’s instructions. DNA was stored at −20 °C until use. The copro PCR was performed according to [20] with some modifications. A sample was only considered positive if *T. solium* DNA was detected in the sample.

### 2.7. Clinical Follow-Up, CT Scans, and Patient Management

A procedure as described for the community-based study in Zambia was followed [11]. For group 1, all patients were seen by a doctor (Figure 2) who performed a neurological examination, while for groups 2 and 3 (Figure 3) only study participants TS POC CC-positive were seen by a study doctor. Appropriate eCRFs were used to record patients’ personal and clinical data according to principles of good clinical practices. After the neurological examination, including past and current medical history, a cerebral CT scan with and without contrast was taken. Contrast was used only if there were no clinical contraindications. If two or more months between the initial TS POC test result and the CT scan elapsed, a new TS POC test was performed using only the strip for CC.

The CT scan results were read independently by the senior project neuroradiologist and an NCC expert. For each CT scan, the number, location, and stage of the lesion were recorded. In the case of a discrepancy, a third independent reviewer adjudicated the CT diagnosis. The final NCC diagnosis was determined according to the 2017 criteria of [8]. These were based on clinical symptoms, neuroimaging, the detection of antibodies and antigens in serum, and epidemiological considerations corresponding to absolute, major, and minor criteria and resulting in the definite and probable diagnosis of NCC. An absolute neuroimaging criterion for NCC was at least one cystic lesion showing the scolex. Major neuroimaging criteria were cystic lesions without scolex, single or multiple rings, or nodular enhancing lesions, multilobulated cystic lesions in the subarachnoid space, and parenchymal brain calcifications. Hydrocephalus or abnormal enhancement of basal leptomeninges were categorized as minor neuroimaging criteria. Four different stages of lesions were defined: vesicular, colloidal vesicular, granular nodular, and nodular calcified. The first three were considered as active NCC lesions and the last was defined as an inactive NCC lesion [8]. 

### 2.8. Patient Treatment and Follow-Up

Treatment with niclosamide (single oral dose, 2 g) or praziquantel (10 mg/kg) (depending on the availability of the drugs) was offered to patients who tested positive for T using the TS POC T or any of the reference tests. Relatives of these patients were also informed and invited for a diagnostic test, and in the case of positivity, treatment was offered. 

Patients who underwent a CT scan were informed about the results during a follow-up appointment, and management procedures as described for the community-based study in Zambia were followed [11]. In short, if indicated, anthelmintic treatment was offered according to national guidelines and Winkler et al. If indicated, anthelmintic treatment with albendazole (15 mg/kg/d) was offered for 10 days, accompanied with corticosteroid treatment (dexamethasone 12 mg to 20 mg/d), starting one day before and ending three days after anthelmintic treatment with half the dosage. Dexamethasone dosage depended on the location and number of cysticerci [21]. Symptomatic NCC cases with active lesions were treated with anthelminthic medication and steroids for two weeks according to the patient’s response. Treatment was tailored to each individual patient. At set time points (after six weeks and again after six months), a follow-up clinical examination was performed. During these time points, follow-up CT scans were offered to monitor treatment success. 

### 2.9. Training

Training and capacity building were cornerstones of the project. Extensive multiple full-day training sessions were organized at the district hospitals to ensure the collection of high-quality data and correct patient management and follow-up. Three-day training sessions were arranged where everyone involved in the project was trained. Sessions on parasite biology, diagnosis, treatment, prevention, and control of the parasite in endemic areas were organized for all project staff members. 

Tailored training was organized for clinicians responsible for recruiting patients, nurses responsible for obtaining informed consent and TS POC testing, and laboratory technicians responsible for blood and stool sample collection and processing. Medical doctors responsible for clinical examinations, including the neurological examination and later patient follow-ups and treatment were trained by the neurology team, contributing to capacity building in the study area. Assessments and follow-up refresher training were planned to assure the correct following of study procedures and good clinical (laboratory) practices and sample collection. Monitoring visits to assess study conduct and compliance on good storage of TS POC tests, samples, and other study material were conducted twice a month. If deviations from standard operating procedures were encountered, corrective measures were initiated.

### 2.10. Data Management

To ensure the collection of high-quality data, all people involved in patient recruitment and performing the POC test were trained and followed up regularly by project members. Non-confidential data were collected electronically on tablets using EpiCollect5 (© 2021 Centre for Genomic Pathogen Surveillance, v4.2.0, https://five.epicollect.net, accessed on 8 July 2021). Specific CRFs were created, and real-time data validation rules were included whenever possible. More details can be found in [11].

Laboratory technicians performing the reference tests were blinded to the TS POC results. Reading of the CT scans was carried out by two independent readers, both blinded to the TS POC and reference tests results. Discrepancies between two readers were identified and solved by a third reader, who was also blinded to the TS POC results.

### 2.11. Sample Size and Statistical Analysis

The sample size for group 1 was calculated to obtain the desired half-width of 5% around the 95% confidence interval for sensitivity and specificity [22]. Assuming a sensitivity of 93%, a specificity of 99% [11], and a prevalence of NCC of 20%, 500 participants were required. To account for losses, 20% were added to this number. The final sample size for group 1 was set at 600.

Since the prevalence of T in the study population was unknown, the sample size for groups 2 and 3 was calculated for different plausible values, ranging between 3% and 10%. Assuming a sensitivity of 82%, a specificity of 81%, and the desired precision of 10%, 1890 participants were required. To account for 5% contingencies, the final sample size was rounded to 2000 patients.

The sensitivity and specificity of the TS POC test for the detection of T, CC, and NCC were determined relative to the reference tests as shown in Table 1, for each of the three study groups. Weighting was used to correct for the study design (a subset of T-CC that is selected for sampling/clinical examination). The positive and negative predictive values were also determined.

For T and CC, the diagnostic sensitivity and specificity of the TS POC test were estimated using a Bayesian approach as described previously [23]. For the evaluation of the TS POC test performance for NCC, sensitivity and specificity were computed by cross-tabulating POC results with those of the final NCC diagnosis. More details can be found in [11].

## 3. Discussion

This trial was designed to provide a rigorous evaluation of TS POC test accuracy for the detection of T, CC, and NCC. The ambitious four-year project was the result of longstanding North–South collaborations and aimed at including 2600 patients of three district hospitals in Tanzania. If the test is validated with success, it may be proposed for commercialization and implementation at district hospitals and community health centers in endemic areas. 

As several sources of bias can arise in diagnostic accuracy studies [24], several measures were taken to assure the internal validity of our study. As unblinded studies may have a higher sensitivity and overall accuracy of results when the person interpreting the reference standard is aware of the index test result [25], diagnostic review bias in the present study was avoided by blinding all laboratory technicians to the TS POC test result. Moreover, as the interpretation of several reference test results for T and CC have inherent test integrity (such as ELISA results, which are read by a spectrophotometer), diagnostic review bias seemed very unlikely for the evaluation of T and CC. On the other hand, clinical examination, and interpretation of CT scans, which were both used for the composite case definition for NCC, are more subjective. Therefore, a structured questionnaire was used to record the results of the neurological examinations to reduce subjectivity in the examinations as much as possible. Moreover, all CT scans were read in detail by two independent readers, and in the case of a disagreement, additionally by a third independent reader, who were all blinded to the TS POC result. 

Stool and serum samples for the reference tests were requested immediately after performing the TS POC test to reduce loss-to-follow-up and to avoid disease progression bias. Although we aimed to perform the CT scans with minimal delay, the CT scans could not always be performed within a reasonable time frame after the index test due to several logistical reasons (such as an inoperative CT scanner and travel planning of the neurology team to be present when CT scans were taken). Since delayed verification may result in biased accuracy estimates [25], an additional TS POC test was performed on patients for whom more than two months had passed between the initial TS POC test and CT scanning. These results were used to assess the extent of disease progression bias.

As disease severity and participant demographics can influence test accuracy, the TS POC test evaluation in the SOLID project included different groups of patients attending the district hospitals for which the test may be used in practice. Moreover, to increase generalizability, three different district hospitals were chosen in the design. To obtain a representative sample of the different studied populations, patients attending the outpatient department (and the mental health ward for patients with specific neurological signs and symptoms) were recruited consecutively for groups 1 and 2, and selecting every tenth patient in group 3. Although several efforts were made to increase the external validity of the study, results may not be generalizable to patients attending local healthcare facilities or other more specialized hospitals, where differences in disease spectrum or other underlying (demographic) factors might be present.

If successful, the tool will be the first POC test contributing to NCC diagnosis which could be made available at the district hospital level. Combined with increased knowledge, training, and capacity building, this project aided staff at the district hospital level to identify, correctly and early, suspected NCC cases. Furthermore, thanks to the training, staff will be able to make the connections between NCC on the one side and epileptic seizures/epilepsy, severe progressive chronic headache, and other neurological signs/symptoms on the other side. Finally, we do hope that patients will be referred to neuroimaging facilities or more specialized hospitals when necessary. This will lead to improved patient management practices, contributing to a positive effect on the patients’ health outcome, and raising awareness of *T. solium*-associated neurological diseases, e.g., epileptic seizures/epilepsy, which bear an enormous burden of the disease, especially in LMIC [26]. NCC is one of the main causes of epileptic seizures/epilepsy in *T. solium*-endemic areas [27] and it is a curable disease. A POC test clearly has the potential to contribute to changing the narrative around epilepsy being a chronic disease with often high mortality in LMIC [26] to a curable disease in *T. solium*-endemic areas. Additionally, a correct diagnosis and knowledge of NCC will potentially stimulate the people’s perception of epilepsy as being a biological disease (e.g., NCC as its cause) rather than a consequence of witchcraft, which is a common belief in the study area [28]. Creating awareness and educating professionals in disease recognition will further lead to an increased number of detected cases. If the TS POC test proves to have acceptable sensitivity and specificity to detect NCC cases, further studies are required to evaluate the TS POC test’s ability to guide clinical decision-making [29].

The TS POC test will be the first portable, easy-to-apply diagnostic test for T available at the district hospital level. This will lead to the early detection and treatment of tapeworm carriers. The latter is crucial to reduce/stop disease transmission as tapeworm carriers are responsible for environmental contamination with infective eggs. These carriers represent a risk not only for themselves but also for their close contacts. Early detection and treatment will ensure a rapid halt to this environmental contamination, reducing the risk of new NCC cases and the infection of pigs, and thus, of perpetuating the life cycle. The training will guide the staff in the treatment and follow-up of tapeworm carriers at the household and community level.

Adequate monitoring and surveillance systems, which are necessary to support the sustainable control of *T. solium*, are currently not implemented in sub-Saharan Africa, largely due to the lack of well-performing tests that are deployable in low-resource settings. Therefore, as well as its diagnostic use, the TS POC may also be used for monitoring and surveillance purposes, as it is affordable, relatively rapid, equipment-free, and user-friendly [14]. For this aim, the TS POC test was also validated for use at the community level in Zambia, which mainly included asymptomatic people [11].

In Tanzania, in contrast to Zambia, the study was conducted primarily by trained hospital staff, and recruitment and testing occurred daily until the target sample size was obtained. To assure the smooth conduct of the study and the correct following of study standard operating procedures, the intensive training of staff involved in the study was essential. In the SOLID project, new digital tools for data collection with barcoding systems were introduced, which might have posed a challenge to members of staff more used to traditional data recording methods. However, the digitalization of data collection was chosen for the SOLID project to be able to monitor study progress by assessing the number of entries uploaded, to assist staff when issues might appear during regular data checks, cleaning, and validation sessions performed by project members, on top of advancing technological knowhow in resource-limited areas. To assure high-quality data collection, correct patient follow-up, and adequate sample collection, processing and storage, the district hospital staff was monitored by project members’ fortnightly visits.

## 4. Conclusions

In conclusion, if the test is successfully validated, the new, rapid, cheap TS POC test might be produced and used in various settings of *T. solium*-endemic areas and assist in diagnosing, monitoring and surveillance, and the epidemiological research of *T. solium* taeniosis/NCC. This will be important for triaging patients for treatment, especially those who suffer from epileptic seizures/epilepsy. In this context, it is important to mention that the development of the WHO NCC management guidelines is in its final stages. Their implementation will rely on validated diagnostic tools, such as the TS POC test. Hence, our study is timely and, depending on the outcome, will contribute to the successful implementation of the WHO NCC management guidelines. Moreover, the intensive training and follow-up visits contributed to capacity building at the district hospital level and increased awareness about the disease and potential management, treatment, and follow-up of affected patients. In turn, this has potential benefits for close contacts, families, and surrounding communities.

## Figures and Tables

**Figure 1 diagnostics-11-01528-f001:**
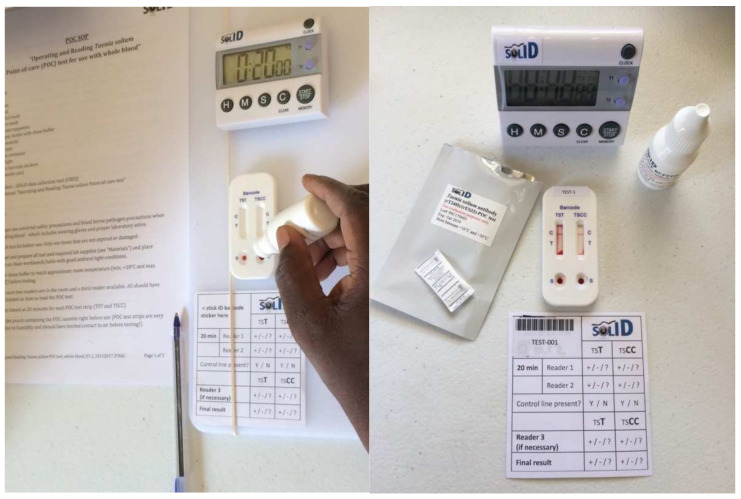
Photos showing the TS POC test, chase buffer, timer set on 20 min, and TS POC result card.

**Figure 2 diagnostics-11-01528-f002:**
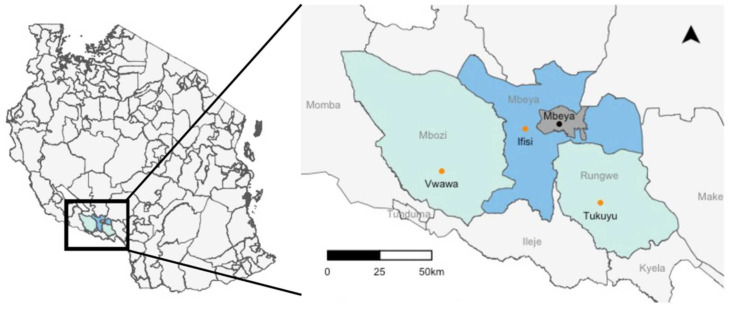
Map of Tanzania showing the study area and a close-up of the district hospitals (orange dots) in Mbozi (Vwawa), rural Mbeya (Ifisi), and Rungwe (Tukuyu), and the CT scan facility in Mbeya (black dot) in the Mbeya and Songwe regions in Tanzania.

**Figure 3 diagnostics-11-01528-f003:**
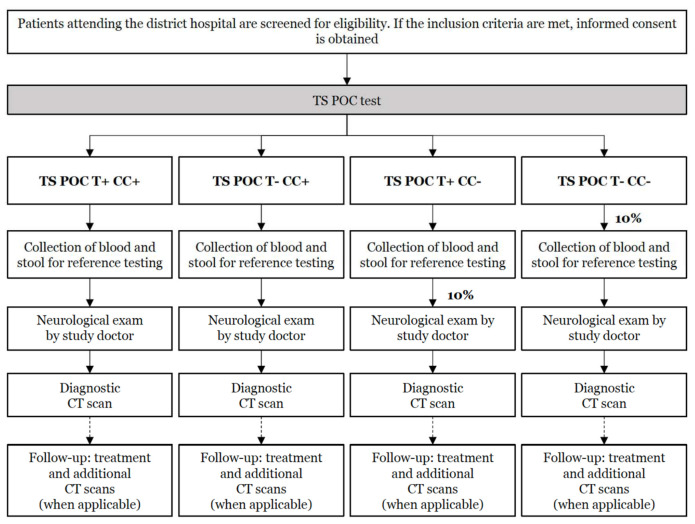
Flow of the diagnostic accuracy study in Tanzania for patients attending the mental health ward or outpatient department of district hospitals with on-going symptoms of severe progressive chronic headache impeding their daily activities without fever and signs of cerebral infection or other obvious causes; and/or a history of one or more epileptic seizures without obvious causes (group 1). In this two-stage design, all patients were tested using the TS POC test (=index test; stage 1), after which all positive, and a subset of negative, patients were selected for different reference tests, clinical examination, and a brain CT scan (stage 2).

**Figure 4 diagnostics-11-01528-f004:**
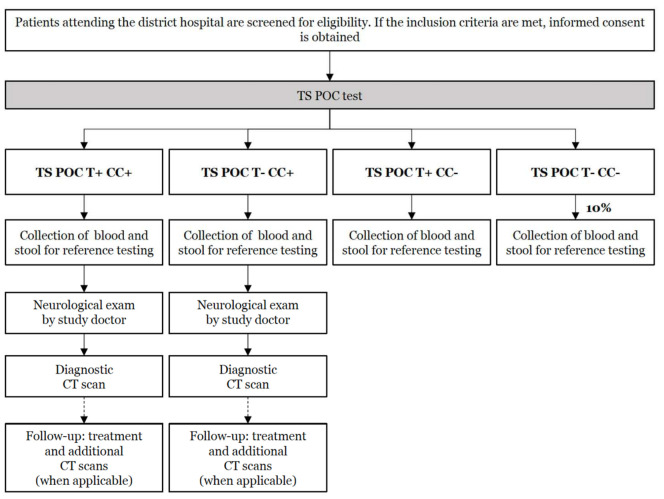
Flow of the diagnostic accuracy study in Tanzania for patients attending the outpatient department of district hospitals with symptoms of intestinal worm infections (group 2) and other symptoms (group 3) that were not specific neurological signs and symptoms. In this two-stage design, all patients were tested using the TS POC test (=index test; stage 1), after which all positive, and a subset of negative, patients were selected for different reference tests. Clinical examination and a brain CT scan (stage 2) were only performed on patients positive for CC on the TS POC test.

**Table 1 diagnostics-11-01528-t001:** Reference tests used to assess the sensitivity and specificity of the TS POC test for the detection of taeniosis, cysticercosis, and neurocysticercosis.

Disease Presentation	TS POC Test Strip	Purpose	Reference Standard
Taeniosis	TS POC T	Diagnosis/treatment Monitoring/surveillance Epidemiological research	copro antigen ELISA (stool ag) copro PCR (stool DNA) rES33 immunoblot (serum ab)
Cysticercosis	TS POC CC	Monitoring/surveillance Epidemiological research	EITB-LLGP (serum ab) rT24H immunoblot (serum ab) serum antigen ELISA (serum ag)
Neurocysticercosis	TS POC CC	Diagnosis/treatment and management Epidemiological research	Composite case definition according to Del Brutto et al.’s criteria [8]

TS POC T: *Taenia solium* point-of-care test taeniosis test strip; ELISA: enzyme-linked immunosorbent assay; PCR: Polymerase Chain Reaction; TS POC CC: *Taenia solium* point-of-care test cysticercosis strip; EITB: enzyme-linked immunoelectrotransfer blot; LLGP: lentil lectin-bound glycoproteins.

## Data Availability

No new data were created or analyzed in this manuscript. Data sharing is not applicable to this article.

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
