# Peer review of "Trial Design of a Prospective Multicenter Diagnostic Accuracy Study of a Point-of-Care Test for the Detection of Taenia solium Taeniosis and Neurocysticercosis in Hospital-Based Settings in Tanzania"

_diagnostics, 2021, doi:10.3390/diagnostics11091528_

Round 1

Reviewer 1 Report

The manuscript by Trevisan et al describes a trial design of a point-of-care test for the parasite Taenia solium. The manuscript is well written, except for a few minor points (see below). However, most of the authors already published a very similar protocol / study design. In this manuscript, the larger study population is described. Since the scope of the journal Diagnostics is to publish “original research articles, reviews, short communications, case reports and interesting images”, I do not see the value of an additional protocol without showing any data. Not even the initial results of the test accuracy were shown, but only referenced to as “data not shown” in the previous protocol published in Diagnostics. The study itself and the point-of-care test are both very important and valuable especially for resource-poor areas, but I am reluctant to suggest publishing of another extremely similar protocol  without showing data. The only difference of the protocol is the setting, number of tests produced and to be performed and the study population.  

If however, the editors consider the protocol for publication, I would ask to consider the following minor points:

Introduction:

  • Last paragraph page 3: “…for the detection of solium T and (N)CC in a resource-poor, highly…” à define T, e.g., put in brackets

Materials and methods:

  • Define the proteins (type) rESS33 & rT24H à in the reference you state, the protein is named TSES33, the other reference does not state a protein, but PCR primers, use the correct references here (e.g., doi: 4269/ajtmh.17-0310; doi: 10.1128/JCM.03260-13)
  • explain better that T-antibodies and CC-antibodies are to detect the parasite or disease
  • 11: the authors mention the sensitivity and specificity of the TSPOC and reference their own previous paper – however, these data were not shown in the paper, at least a summary of this should be shown and not just said “unpublished data”

Reviewer 2 Report

The paper describes an interesting study design, the simulatneous testing for taeniosis and (neuro-)cysticercosis in three patient cohorts in Tanzania.

While the study design seems approriate and follows the need for fast and reliable testing in resource-poor settings, the paper does not present any study results as - apparently - the study has not yet been performed. I strongly suggest to validate the mentioned test with 10 patients, run a pilot study with 10-30 pateints and present these limited results alongside the study design, and wait for the finalized WHO NCC management guidelines.

A bit confusing, the sentences are written in past tense as if the study had been performed already.

Round 2

Reviewer 1 Report

Thank you for clarifying some aspects.

Reviewer 2 Report

Thank you for the clarification.